



# Dissolved organic matter fosters core mercury-methylating microbiome for methylmercury production in paddy soils

Qiang Pu[1], Bo Meng[1], Jen-How Huang[1], Kun Zhang[1,2], Jiang Liu[1], Yurong Liu[3], Mahmoud A. Abdelhafiz[1,2,4], Xinbin Feng[1,2]

[1]State Key Laboratory of Environmental Geochemistry, Institute of Geochemistry, Chinese Academy of Sciences, Guiyang, 550081, China
[2]University of Chinese Academy of Sciences, Beijing, 100049, China
[3]State Key Laboratory of Agricultural Microbiology and College of Resources and Environment, Huazhong Agricultural University, Wuhan 430070, China
[4]Geology Department, Faculty of Science, Al-Azhar University, Assiut 71524, Egypt

*Correspondence to:* Bo Meng (mengbo@mail.gyig.ac.cn)



**Abstract.** Methylmercury (MeHg), accumulated in rice grain, is highly toxic for human. Its production is largely driven by microbial methylation in paddy soils; however, dissolved organic matter (DOM) represents a hotspot for soil biogeochemistry, resulting in MeHg production, remain poorly understood. Here, we conducted *hgcA* gene sequencing and genome-resolved metagenomic analysis to identify core Hg-methylating microbiome and investigate the effect of DOM on core Hg-methylating microbiome in paddy soils across a Hg contamination gradient. In general, the Hg-methylating microbial communities varied largely with the degree of Hg contamination in soils. Surprisingly, a core Hg-methylating microbiome was identified exclusively associated with MeHg concentration. The partial Mantel test revealed strong linkages among core Hg-methylating microbiome composition, DOM and MeHg concentration. Structural equation model further indicated that core Hg-methylating microbiome composition significantly impacted soil MeHg concentration (accounting for 89%); while DOM was crucial in determining core Hg-methylating microbiome composition (65%). These results suggested that DOM regulates MeHg production by altering the composition of core Hg-methylating microbiome. The presence of various genes associated with carbon metabolism in the metagenome-assembled genome of core Hg-methylating microorganisms suggests that different DOMs stimulate the activity of core Hg-methylating microorganisms to methylate Hg, which was confirmed by pure incubation experiment with *Geobacter sulfurreducens* PCA (core Hg-methylating microorganism) amended with natural DOM solution extracted from investigated soils. Overall, DOM simultaneously changes core Hg-methylating microbiome composition and functional activity and thus enhances MeHg production in paddy soils.

**Keywords.** Rice paddy; Mercury methylator; Methylmercury formation; Core microbiome



32 **Graphical abstract**

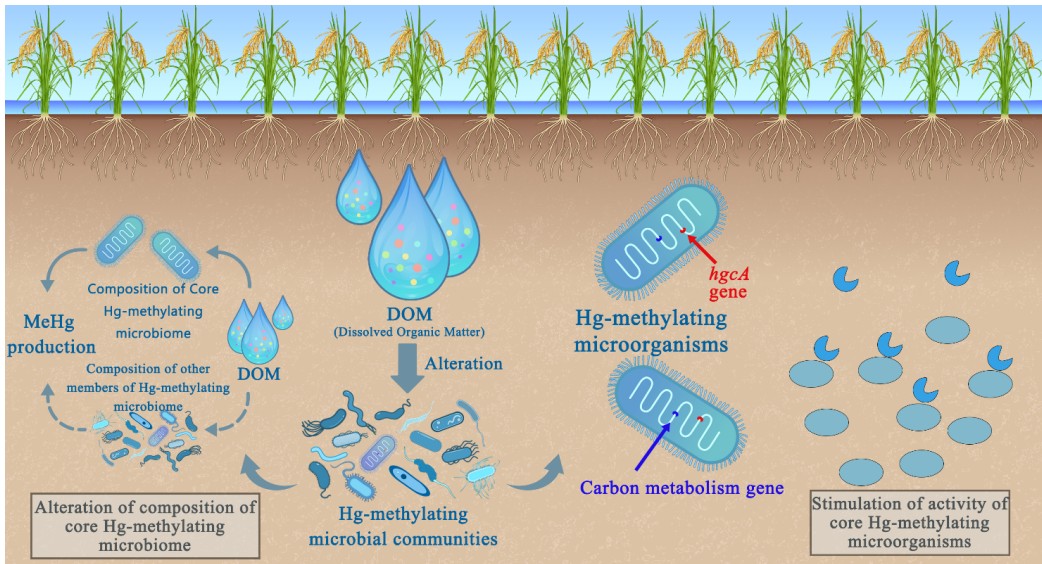

33

34



## 1 Introduction

Mercury (Hg) is a toxic contaminant since it can be transformed into neurotoxic methylmercury (MeHg) and biomagnified in food chains (Driscoll et al., 2013). Human exposure to MeHg can cause neurocognitive deficits and cardiovascular effects (Oulhote et al., 2017; Roman et al., 2011). It is generally accepted that seafood consumption is the major route of exposure to MeHg in humans (Schartup et al., 2019). Recent studies have demonstrated that rice consumption is another important route of human exposure to MeHg (Feng et al., 2008), as 3.5 billion individuals relying on rice as principal dietary component (Muthayya et al., 2014). The accumulation of MeHg in rice is mostly attributed to microbial methylation of inorganic Hg in paddy soils (Meng et al., 2011). *In-situ* methylation and demethylation are deemed to be important processes controlling the net MeHg concentration in environments (Barkay and Gu, 2022; Helmrich et al., 2021; Li and Cai, 2012). Our recent study showed that Hg transformation processes, such as methylation, demethylation, oxidation, and reduction, occurred simultaneously in paddy soils, with Hg methylation being the most active (Liu et al., 2023). Therefore, paddy soil is a typical "hotspot" for Hg methylation, which is mainly a biotic process mediated by many abiotic factors, such as Hg bioavailability and redox conditions (Li and Cai, 2012). The diversity and activity of Hg-methylating microorganisms in paddy soils controls MeHg production (Gilmour et al., 2013; Liu et al., 2018b). However, among the various Hg-methylating microorganisms currently known, the core microbiome controlling MeHg production and its interaction with environmental variables in paddy soils have yet to be identified.

Physicochemical factors in soils, such as organic matter, pH, salinity, redox potential, iron, and sulfur, have been shown to regulate the activity of Hg-methylating microorganisms and play an important role in controlling MeHg production in rice fields (Ullrich et al., 2001). Among the different variables, soil organic matter, which is ubiquitous in paddy soils (Li et al., 2018), play a vital role in Hg methylation (Yin et al., 2013). Dissolved organic matter (DOM), the most mobile organic matter fraction, increases MeHg production under sulfidic conditions (Graham et al., 2012). DOM increases microbial Hg bioavailability for methylation by stabilizing $\beta$-HgS(s) nanoparticles to prevent aggregation. In addition, Hg speciation in Hg-contaminated paddy soils was found to be predominantly regulated by organic matter (Liu et al., 2022), and the high bioavailability of DOM-bound Hg in rice paddies contributed to an increase in MeHg production (Liu et al., 2022). In contrast, other studies reported that DOM had a high affinity for Hg compounds (Skyllberg et al., 2006), suppressing MeHg production due to strong Hg-DOM complexation (Schartup et al., 2015). As a result, the role of paddy soil DOM on Hg methylation remains elusive. Our recent study showed a significant and strong relationship between MeHg production and lower molecular weight DOM in paddy soils collected from major rice-producing areas across China (Abdelhafiz et al., 2023). Given paddy soil DOM's significant chemodiversity (Li et al., 2018), it is reasonable to hypothesize that the effect of DOM on MeHg production cannot be assessed solely based on Hg speciation and bioavailability, suggesting that other factors also play a role in MeHg production.

MeHg production is controlled by the synergy of Hg bioavailability and Hg-methylation capacity (Peterson et al., 2023), indicating that Hg-methylating microbial communities may also play an important role in DOM-regulated MeHg production. Concentration and composition of DOM have been shown to regulate MeHg production via alteration of the composition of the soil microbial community (Fagervold et al., 2014; Hu et al., 2021; Oloo et al., 2016). However, the core Hg-methylating microorganisms were not identified within these studies. Zhao et al. (2017) reported that two model Hg methylators exhibited an opposite response to DOM at the strain level. Therefore, we hypothesized that DOM fosters a core Hg-methylating microbiome that regulates MeHg production, since the core microbiome has a pivotal role in the functioning of ecosystems (Banerjee et al., 2018; Chen et al., 2019; Xun et al., 2021).



Thus, an attempt was made within this study to verify the crucial role of DOM in fostering the core Hg-methylating
microbiome for MeHg production by (1) identifying the core Hg-methylating microbiome in paddy soils across a gradient
of Hg contamination, (2) quantifying the relevance of DOM to core Hg-methylating microbiome and MeHg production
in paddy soils compared with other soil physicochemical parameters, and (3) elucidating the mechanism of core Hg-
methylating microorganisms in response to different DOMs. These results broaden our understanding of DOM as the
prominent factor in altering Hg-methylating microbial communities and highlight the contribution of the core Hg-
methylating microbiome to MeHg production in paddy soils.
**2 Materials and methods**
**2.1 Soil sampling and physico-chemical analysis**
Two field sampling campaigns were conducted in September 2020 and August 2022 in this study. Specifically, paddy
fields from an abandoned Hg mining area (Sikeng, SK), an artisanal Hg smelting area (Gouxi, GX), and a regional
background area (Huaxi, HX) in Guizhou Province, SW-China, were selected in September 2020 (Table S1, S1- S27). In
each study area (SK, GX, and HX), nine sampling sites were randomly selected. Similarly, additional 19 sampling sites
from the rice producing areas in 12 provinces of China were selected in August 2022 (Table S1, S28-S46). At each site,
one rice paddy field was randomly selected. Paddy soil was taken from the root zone (10-20 cm deep) and comprised a
composite of three subsamples from the same paddy field. A total of 46 soil samples were obtained in this study to
represent different Hg contamination levels and bioavailability, net MeHg production, DOM concentration and
composition, soil microbial community composition and structure, and other physicochemical characteristics. Soil
samples were collected in the sterile PP bottles (Nalgene®, Thermo Fisher, USA) without any headspace, immediately
shipped back to the laboratory on ice packs (~4°C) and divided into two subsamples before use. One subsample was
stored at -20°C for microbial analysis, and the other was stored at 4°C for the analysis of soil physicochemical properties.
The details on the measurements of soil pH, total carbon and total nitrogen, Hg species (water-soluble Hg, total Hg (THg),
and MeHg), $SO_4^{2-}$ and $NO_3^-$ (measured as water-soluble $SO_4^{2-}$ and $NO_3^-$), DOM concentration (measured as water-
soluble dissolved organic carbon), DOM composition (measured as optical properties of DOM), iron and sulfur (measured
as $Fe^{2+}$ and $S^{2-}$ in soil pore water) are presented in Supplement Text S1. It should be noted that $Fe^{2+}$ and $S^{2-}$data were
limited to soil samples obtained in August 2022.
**2.2 Soil DNA extraction and analysis of Hg-methylating microbial communities**
The MP Biomedicals FastDNA Spin Kit was used to extract soil DNA according to the manufacturer's instructions. Soil
Hg-methylating microbial communities were characterized by Illumina MiSeq sequencing of the *hgcA* gene using the
primer pair ORNL-HgcAB-uni-F (5'-AAYGTCTGGTGYGCNGCVGG-3′) and the reverse primer ORNL-HgcAB-uni-
32R (5'-CAGGCNCCGCAYTCSATRCA-3′) (Gionfriddo et al., 2020). Amplicons were equimolarly mixed, and
sequenced using the Illumina MiSeq instrument (Illumina Inc., San Diego) in 2×300 bp mode. Poor-quality reads, adapters
and primers were trimmed with SICKLE and CUTADAPT (Joshi and Fass, 2011; Martin, 2011). USEARCH (version
8.0) was used to truncate, dereplicate, sort and remove singletons (Edgar, 2013). The set of sequences obtained was
clustered at a 60% similarity cutoff with cd-hit-est (Fu et al., 2012). Using USEARCH (version 8.0), the sequences were
then mapped to the resulting clusters' representative sequences to build a count table. The sequences were annotated with
amino acid sequences from Hg-MATE-Db (V1.01142021) (Gionfriddo et al., 2021) by using a Hidden Markov Model
(HMM) based on HMMER (Eddy, 2011). In addition, the abundance of the Hg-methylating gene (*hgcA*) was quantified
in an Applied Biosystem 7500. The quantification of the *hgcA* gene is provided in Text S2.



**2.3 Shotgun metagenomic analysis via Illumina sequencing**

The DNA from nine randomly chosen paddy fields at each site in September 2020 was equimolarly mixed together to obtain >1 μg of DNA for shotgun metagenomic sequencing. For paddy soils collected in August 2022, three replicates of each sample were utilized to ensure sufficient quantity and quality of DNA for metagenomic sequencing. In total, 22 samples were used for metagenomic analysis. Sequencing was performed with an Illumina HiSeq 2500 system (Illumina Corp., USA).

The detection and taxonomic identification of the *hgcAB* gene was performed with marky-coco (Capo et al., 2023). The metagenomic sequences were trimmed to eliminate low-quality reads using fastp with the following parameters: -q 30 -l 25 --detect_adapter_for_pe --trim_poly_g --trim_poly_x (Chen et al., 2018). These high-quality reads were then assembled into contigs using megahit 1.1.2 with default settings (Li et al., 2016). The annotation of the contigs for prokaryotic protein-coding gene prediction was conducted using prodigal 2.6.3 (Hyatt et al., 2010). To search for *hgc* homologs, a profile of HMM derived from Hg-MATE.db.v1 was applied to amino acid FASTA file generated from each assembly with the function hmmsearch from HMMER 3.2.1 (Finn et al., 2011). To eliminate paralogs of *hgcA*, we removed the sequences without the conserved putative cap helix motif [N(V/I)WCA(A/G)GK] reported previously (Parks et al., 2013). We further filtered the sequences by retaining only sequences with more than four transmembrane domains as identified by TMHMM (v.2.0) (Krogh et al., 2001). Finally, the obtained contigs with *hgcA* homologs were classified taxonomically following a previously described method (Zhang et al., 2023). In addition, to estimate the relative abundance of the *hgcA* gene, metagenomic reads were mapped to representative genomes of the *hgcA* dataset using Bowtie2 (Capo et al., 2023). The relative abundances of each gene were calculated by normalizing the total length of successfully mapped reads by gene length and the total number of reads in the metagenome.

Contigs ≥ 1000 bp were used to carry out binning analysis with the MetaWRAP pipeline (v1.3.2) (Uritskiy et al., 2018). The quality of reconstructed metagenome-assembled genomes (MAGs) was assessed using CheckM (Parks et al., 2015). High-quality MAGs (completeness ≥ 90% and contamination ≤ 10%) were used to detect *hgcA* homologs, and taxonomy of these retrieved MAGs was conducted using GTDB-tk (v2.1.0) with its reference database (version release_207V2) (Parks et al., 2022). To explore what fractions of DOM can be metabolized by core Hg-methylating microorganisms, core Hg-methylating microbial-associated MAGs were mapped to the protein sequence of the Kyoto Encyclopedia of Genes and Genomes (KEGG) database using eggNOG mapper (Huerta-Cepas et al., 2017).

**2.4 Pure incubation of *Geobacter sulfurreducens* PCA with different DOMs**

To validate that different concentrations and molecular weights of DOM stimulate the activity of core Hg-methylating microorganisms, we incubated *Geobacter sulfurreducens* PCA (*G. sulfurreducens* PCA, core Hg-methylating microorganism in this study) with $Hg^{2+}$, natural DOM solution extracted from NMS, MMS, and HMS, respectively. More details on the descriptions for the pure incubation experiment can be found in Text S3.

**2.5 Statistical analysis**

Statistical analysis was conducted with SPSS 27 (SPSS, Chicago, IL), AMOS (SPSS, Chicago, IL), and R platform (version 3.6.1). All statistical tests were considered significant at $p < 0.05$. The Mann-Whitney U test statistic was used to compare microbial alpha diversity among all samples. The overall pattern of Hg-methylating microbial communities was determined by analysing dissimilarity matrices using Bray-Curtis distance and compared among different Hg polluted soils using principal coordinates analysis (PCoA) and Adonis with the "ade4" and "vegan" packages (Dray and Dufour, 2007; Oksanen et al., 2017). To determine the relationship between THg and MeHg, Spearman correlation was performed using "ggpubr" and visualized using "ggplot2" packages (Kassambara, 2018; Wickham, 2009). Variation partitioning



analysis was performed using "vegan" package (Oksanen et al., 2017). The major predictors of Hg-methylating microbial
communities and their significance were identified using random forest analysis with "randomForest", "rfPermute" and
"A3" packages (Archer, 2018; Fortmann-Roe, 2015; Liaw and Wiener, 2002). To investigate the co-occurrence patterns
among microbial taxa related to MeHg production, co-occurrence networks were established in the R platform using
"psych" package, and visualized in Gephi 0.9.2 based on strong (Spearman's r > 0.8) and significant ($p < 0.01$)
correlations (De Caceres and Legendre, 2009). The modules in Hg-methylating microbial network were identified using
default parameters from Gephi. To explore the relationship between the modules and environmental parameters, we
correlated dissimilarities of bacterial composition in core Hg-methylating microbiome with those of environmental factors
as previously described (Sunagawa et al., 2015). The structural equation model (SEM) was conducted using AMOS 28
to evaluate the impacts of DOM and core Hg-methylating microbiome on MeHg production. A *prior* model was
established based on the known relationships among drivers impacting MeHg production (Fig. S1). We further calculated
the contribution of diverse ecological parameters, especially DOM, to core Hg-methylating microbiome and the
contribution of core Hg-methylating microbiome to MeHg production as previously described (Tao et al., 2015).
**3 Results**
**3.1 Mercury production in paddy soils**
THg concentrations in paddy soils ranged from 0.03 to 1079.75 μg/g dw (Table S1). As reported in our previous study,
dividing paddy soils by THg concentration rather than sampling sites facilitates a comprehensive investigation of the key
factors influencing Hg methylation (Abdelhafiz et al., 2023). Therefore, the paddy soils in this study were divided into
three categories according to THg concentration: non-Hg contaminated soils (NMS, with average levels of 0.24 ± 0.18
μg/g dw, n=23), moderate Hg-contaminated soils (MMS, 18.28 ± 6.77 μg/g dw, n=13), and high Hg-contaminated soils
(HMS, 637.79 ± 160.93 μg/g dw, n=10). Furthermore, statistically significant differences in DOM concentrations
(reflected by DOC concentration) and DOM composition (reflected by $S_R$ of DOM) were found in NMS, MMS and HMS
(Table S2). However, no discernible differences in physicochemical properties (e.g., pH, $S^{2-}$, $SO_4^{2-}$, $NO_3^-$, TN, TC, $Fe^{2+}$)
were observed in NMS, MMS and HMS (Table S3).
In this study, we found MeHg concentration in paddy soils in the order of HMS (5.01 ± 0.77 ng/g dw, n=10) >> MMS
(2.54 ± 0.72 ng/g dw, n=13) > NMS (0.76 ± 0.25 ng/g dw, n=23) (Fig. S2). Accordingly, a positive relationship was
observed between total Hg and MeHg in different paddy soils (Fig. S3).
**3.2 Core mercury-methylating microbiome as predictors of MeHg production in paddy soils**
Random forest result revealed that *hgcA* gene abundance, DOM concentration, DOM composition, water-soluble Hg,
$Fe^{2+}$, and $S^{2-}$ were significantly ($p < 0.05$) associated with MeHg concentration (Fig. S4), with the *hgcA* gene as the
strongest predictor. The *hgcA* gene-base taxonomic profiles of Hg-methylating microbial communities reveal the vast
composition of paddy soils using the *hgcA* gene sequencing approach (Fig. 1a). Such observations were additionally
supported by (1) the Chao1 index revealing the diversity of Hg-methylating microorganisms in the order of MMS (312.57
± 44.73) > NMS (268.47 ± 81.85) > HMS (187.08 ± 131.62) ($p < 0.05$; Fig. 1b) and (2) the divergent patterns of Hg-
methylating microbial communities in paddy soils ($p < 0.01$; Fig. 1c). The shotgun metagenomics results were consistent
in detecting Hg-methylating microbial community composition and structure (Fig. S5). *Proteobacteria*, *Acidobacteria*,
and *Chloroflexi* were the most abundant phyla in different paddy soils detected by both sequencing strategies. In summary,
using both *hgcA* gene sequencing and metagenomic data, a significant difference in Hg-methylating microbial community
structure and diversity was observed in paddy soils.



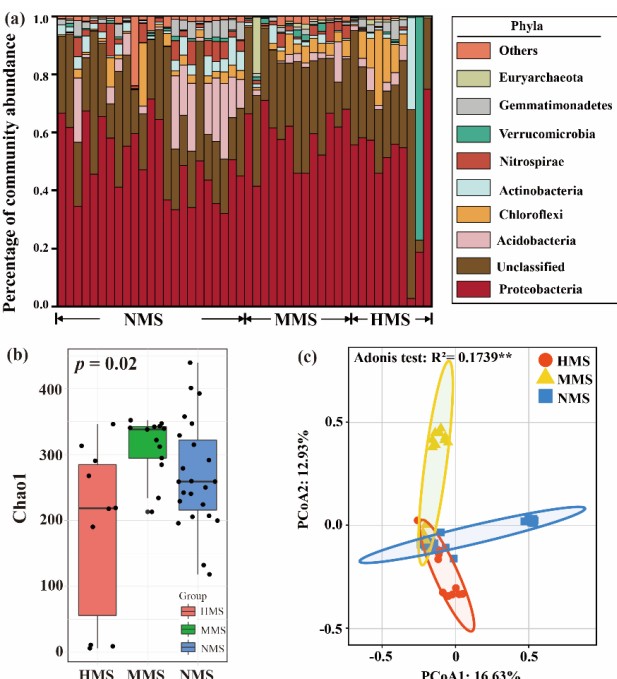

**Figure 1: Taxonomic profiles of Hg-methylating microbial communities in paddy soils. (a)** Microbial community composition in differently polluted paddy soils. Phyla with low abundance phyla grouped together under "other phyla". **(b)** Microbial diversity (based on the Chao1 index) in differently polluted paddy soils. **(c)** Principal coordinates analysis (PCoA) based on Bray-curtis distance showing the overall pattern of Hg-methylating microbial communities in differently polluted paddy soils. NMS, non-Hg polluted paddy soils (n = 23); MMS, moderate Hg-polluted paddy soils (n = 13); HMS, high Hg-polluted paddy soils (n = 10).

Network analysis captured six, eleven, and eleven modules (modularity index > 0.55) in NMS, MMS, and HMS, respectively (Fig. 2a, Table S4). Among all modules, Hg-methylating microorganisms in Module1 in NMS, MMS and HMS were identified as core Hg-methylating microbiome based on their (1) higher connections to other modules and (2) higher abundance in total Hg-methylating microbial community (Table S5). Importantly, the core Hg-methylating microbiome was identified as an important bacterial taxon of soil MeHg concentration (Fig. 2b). Further analysis of the core Hg-methylating microbiome composition revealed diverse core Hg-methylating microorganisms in paddy soils, with the dominant Hg-methylating genera being *Geobacter*, *Desulfuromonas*, and *Methanoregular* (Fig. 2c).




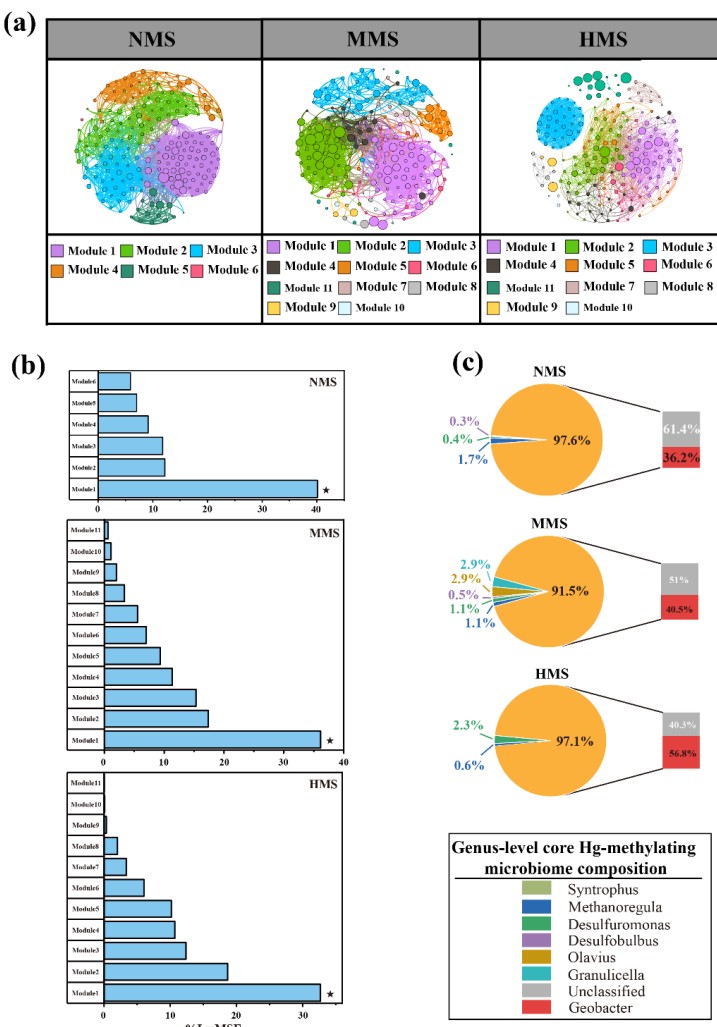

205

**Figure 2: Core Hg-methylating microbiome in paddy soils. (a)** Co-occurrence network of Hg-methylating microbial community in differently polluted paddy soils. Each node represents one OTU. The node size is proportional to the relative abundance of OTUs. **(b)** Predictors of the MeHg production in differently polluted paddy soils based on Random Forest analysis. Only predictors with significant effects are labeled asterisks. **(c)** Core Hg-methylating microbiome composition at genus level in differently polluted paddy soils. NMS, non-Hg polluted paddy soils (n = 23); MMS, moderate Hg-polluted paddy soils (n = 13); HMS, high Hg-polluted paddy soils (n = 10).

### 3.3 Dissolved organic matter as indicators of core mercury-methylating microbiome composition in paddy soils

Based on analysis of correlations, the results showed that there were significant correlations between core Hg-methylating microbiome composition, MeHg concentration, DOM concentration, DOM composition, water-soluble Hg, soil $S^{2-}$ and $Fe^{2+}$ (Fig. 3). Among all parameters, DOM is the most important factor influencing the composition of core Hg-methylating microbiome. This was supported by DOM explaining the most to core Hg-methylating microbiome composition (Fig. S6). Random forest result also showed that DOM concentration and composition were the most important predictors of the composition of core Hg-methylating microbiome (Fig. S7).



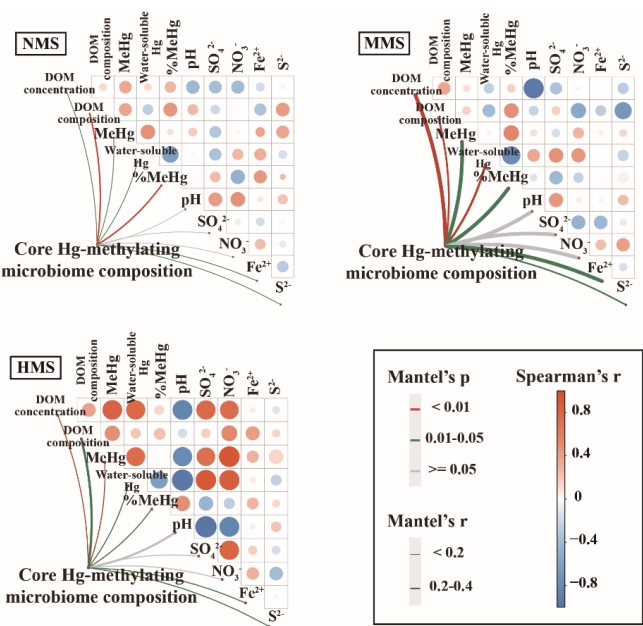

**Figure 3. Pairwise comparisons of environmental factors and community taxonomic composition in core Hg-methylating microbiome in differently polluted paddy soils.** NMS, non-Hg polluted paddy soils; MMS, moderate Hg-polluted paddy soils; HMS, high Hg-polluted paddy soils.

Overall, the core Hg-methylating microbiome composition was interactively affected by DOM ($\lambda = 0.86$, $p < 0.001$), redox conditions ($\lambda = 0.53$, $p < 0.01$), and Hg bioavailability ($\lambda = 0.33$, $p < 0.01$), and alteration of the core Hg-methylating microbiome composition significantly regulated soil MeHg concentration ($\lambda = 0.84$, $p < 0.001$) (Fig. 4).

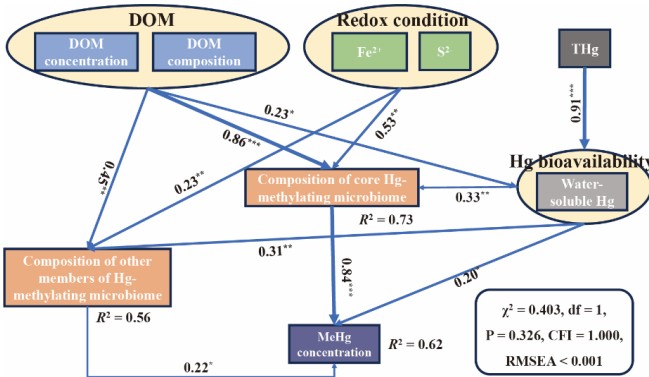

**Figure 4. Structural equation models showing the effects of DOM, redox conditions, and Hg bioavailability on MeHg production.** NMDS1 values of the NMDS analysis were used for the representation of DOM and Redox condition in the SEMs. Numbers adjacent to arrows are standardized path coefficients, and numbers in brackets denote p values. 'Statistically nonsignificant' results are not shown in the figure. $R^2$ denotes the proportion of variance explained.

### 3.4 Dissolved organic matter stimulates activity of core mercury-methylating microorganism enhancing methylmercury production in paddy soils

The results of metagenomic-binning revealed that three core Hg-methylating microbial-associated metagenome-assembled genomes (MAGs, completeness $\geq 90\%$ and contamination $\leq 10\%$) carried different carbon utilization genes (*ackA*, *sdhA*, or *ppdK* gene) (Fig. 5), which are responsible for acetate kinase, succinate dehydrogenase, pyruvate and



orthophosphate dikinase. These results indicated that the low-molecular weight DOMs in soil selectively stimulate the
activity of core Hg-methylating microorganism that preferentially utilize them for metabolism, leading to the increase of
MeHg concentration.

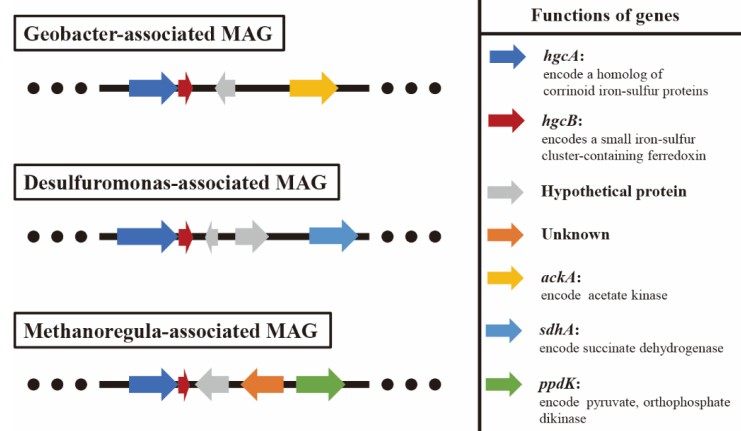

**Figure 5. Analysis of the genetic context of *hgcA* gene and genes involved in carbon metabolism in core Hg-methylating**
**microbial-associated MAGs.** The extents and directions of genes are shown by arrows labeled with gene names.
To validate this hypothesis, *Geobacter sulfurreducens* PCA, core Hg-methylating microorganism identified in this
study, was incubated with HgCl$_2$ and various DOM solutions extracted from investigated paddy soils. The results showed
distinct patterns in MeHg production (Fig. 6), confirming that different concentration of low-molecular weight DOMs
significantly regulates MeHg production by influencing the activity of core Hg-methylating microorganisms.

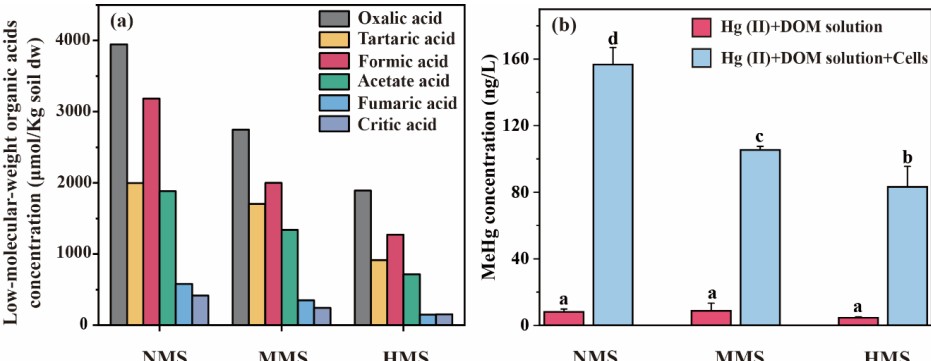

**Figure 6. Effect of natural DOM solution extracted from paddy soils on MeHg production by core Hg methylator (*Geobacter***
***sulfurreducens* PCA). (a)** The concentration of low-molecular-weight organic acids in paddy soils from non-Hg polluted soils (NMS),
moderate Hg-polluted soils (MMS) and high Hg-polluted soils (HMS). **(b)** MeHg concentration by *G. sulfurreducens* PCA. Data (n =
3) are presented as mean value ± SD, with error bars representing standard deviations. Significant differences among different
treatments were tested with Tukey's honest significance test; different lowercase letters in each bar indicate significant differences
among treatments ($p$ < 0.05).
**4 Discussion**
Our study found that MeHg concentration was strongly linked to *hgcA* gene abundance even compared to abiotic factors,
which suggested that MeHg production is a microbially-mediated process (Parks et al., 2013; Podar et al., 2015). Our
study further revealed that although there are significant differences in the Hg-methylating microbial communities in
different polluted paddy soils, they all have a core Hg-methylating microbiome, which plays a more important role than



other Hg methylators in regulating MeHg production. As illustrated by a previous study, the major module (also known
as the core microbiome) in microbial community network contributes to the stability of soil microbiome, enhancing its
resistance to climate changes and nutrient fertilization (Jiao et al., 2022). These findings establish the presence of a major
module contributing exclusively to Hg methylation in paddy soils, although there are many more Hg-methylating
microorganisms present. In fact, microorganisms containing the *hgcA* gene are able to methylate Hg, but this does not
mean that they are automatically active in Hg methylation.
The SEM analysis result indicated that although redox conditions and Hg bioavailability significantly affected the
composition of core Hg-methylating microbiome, their contribution to the composition of core Hg-methylating
microbiome was less and weaker than that of DOM. Specifically, the contributions of Hg bioavailability and redox
conditions to the core Hg-methylating microbiome composition are 10% and 25%, respectively, which are much lower
than that of DOM (65%). The explanation for this phenomenon may be that (1) the soil collected in the paddy field during
the flooding period is in an anaerobic state, so the selection of redox conditions on core mercury-methylating
microorganisms is weakened; (2) Hg is a toxic element to microorganisms and is usually not involved in microbial
metabolism (Wang et al., 2020). Environmental Hg may usually induce the persistence of microorganisms. Therefore,
long-term Hg contamination often elevates the abundance of specific microbial taxa capable of Hg tolerance (Frossard et
al., 2018); (3) DOM, an important carbon source and nutrient in nature, is involved in microbial respiration and
metabolism (Kujawinski, 2011). Therefore, the concentration and composition of DOM contributed significantly to core
mercury-methylating microbiome. Although DOM, redox conditions and Hg bioavailability are capable of influencing
microbial Hg methylation (Liu et al., 2018a; Xu et al., 2021), our results manifest for the first time that DOM plays a
more prominent role in MeHg production than Hg bioavailability and redox conditions by altering core Hg-methylating
microbiome composition.
Our study found that *Geobacter*, *Desulfuromonas*, and *Methanoregular* are core Hg-methylating microorganisms in
paddy soils. Previous studies confirmed that *Geobacter* and *Desulfuromonas* have the capability for Hg methylation
(Bravo et al., 2018; Liu et al., 2018b). In addition, *Methanoregular* spp., as methanogenic archaea, show potential for Hg
methylation (Jones et al., 2019). However, our study highlights that their role in Hg methylation in paddy soils was much
higher than previously thought. A subsequent binning approach was performed to identify these three core Hg-
methylating microbial-associated MAGs, and the results showed that these MAGs contained the *ackA*, *sdhA*, or *ppdK*
genes. This result suggests that different DOMs can activate different Hg-methylating microorganisms that utilize them
for metabolism, thereby providing evidence that DOM can alter core Hg-methylating microbiome composition in paddy
soils. In summary, these three core Hg-methylating microbial-associated MAGs carry different carbon metabolism genes,
further supporting our results that low-molecular-weight DOMs in paddy soils stimulate the activity of Hg-methylating
microorganisms, simultaneously upregulating *hgcA* gene expression.
Our study observed the presence of various DOMs (oxalic acid, tartaric acid, formic acid, acetate acid, fumaric acid,
and critic acid) in paddy soils, indicating that the utilization of different DOMs by Hg-methylating microorganisms can
stimulate the growth of Hg-methylating microorganisms, thereby forming core Hg-methylating microbiome. For example,
*Geobacter sulfurreducens* PCA and *Desulfovibrio desulphuricans* ND132 preferentially used acetate/fumarate and
pyruvate/fumarate, respectively (Hu et al., 2013). *Geobacter anodireducens* SD-1 utilized acetate and citrate favourably
(Liu et al., 2018b). *Methanocella arvoryzae* MRE50(T) thrived on $H_2/CO_2$ and formate (Sakai et al., 2010).
*Methanosarcina acetivorans* spp. selectively utilized acetate and methanol (Schöne et al., 2022). Pure incubation of
*Geobacter sulfurreducens* PCA (core Hg-methylating microorganism identified in our paddy soils) further revealed that
different concentration of low-molecular weights DOM solution extracted from natural paddy soils obtained from NMS,



MMS and HMS had significant effects on MeHg concentration. These results suggest that DOM indeed stimulate the
activity of core Hg-methylating microorganisms for MeHg production.
The present study revealed that different concentration and composition of DOM have been known to shift microbial
MeHg production. In the case of Hg methylation, DOM complexation was shown to alter the bioavailability of Hg for
methylation (Dong et al., 2011; Jiang et al., 2018; Liu et al., 2022). Here, great emphasis was placed on the effects of
interaction between DOM and core Hg-methylating microbiome on Hg methylation. Human activities and climate
changes significantly change the DOM concentration and composition (e.g., molecular weight, aromaticity, and
bioactivity) in different environmental compartments (Xenopoulos et al., 2021). Over the long term, more stable DOM
would be scattered in the form of black carbon globally due to incomplete fuel and biomass combustion (Qi et al., 2020).
In parallel, DOM could be simpler, smaller, and potentially more reactive in aquatic ecosystems (Xenopoulos et al., 2021).
Thus, the knowledge gained within this study suggests that the variation in DOM quality as a consequence of human
activities would remarkably alter MeHg production rates in different environmental compartments. Nonetheless, the
current state of knowledge does not allow us to know whether such changes would increase or decrease Hg ecotoxicity
in the environment. Therefore, further in-depth studies of the coupling between carbon and Hg are indispensable, which
are able to deliver more accurate assessments of the environmental and health impacts of Hg, especially after the
implementation of the Minamata Convention.
**5 Conclusions**
This study provides novel evidence that DOM significantly influences MeHg production via changes in the composition
and functional activity of the core Hg-methylating microbiome. Although DOM regulates the composition of other
members of the Hg-methylating microbiome, it showed little contribution to MeHg production. Comparatively, DOM
accelerated MeHg production by altering the composition of core Hg-methylating microbiome. Metagenomic-binning
and pure incubation experiment confirmed that different concentration of low-molecular weights DOM stimulates the
activity of core Hg-methylating microorganism, thereby promoting MeHg production. As a result, DOM may also affect
Hg methylation mainly through altering core Hg-methylating microbiome composition and boosting the growth of core
Hg-methylating microorganisms. Our findings suggest that, the changes in DOM concentration and composition due to
human activities and climate change may ultimately have an impact on methylmercury formation and food security.
*Data Availability.* The raw reads of *hgcA* gene amplicon sequencing have been deposited in the NCBI SRA under
accession number PRJNA847325 and PRJNA972506. Shotgun metagenomic sequencing have been deposited in the



NCBI SRA under accession number PRJNA848068 and PRJNA972502. Other datasets generated during the current study
are available from the corresponding author upon reasonable request.
*Author Contributions.* The study was designed by QP, BM, and XBF. QP, JL and YRL conducted the sampling, performed
the DNA extraction and the bioinformatic analyses. JHF, KZ and MA performed the geochemical analyses. The
manuscript was written by QP and BM, with assistance and input from co-authors.
*Competing Interests.* The contact author has declared that none of the authors has any competing interests.
*Acknowledgments.* We appreciate Prof. Alexandre J. Poulain (University of Ottawa, Canada) for his valuable advice on
manuscript writing. We also appreciate Prof. Tao Jiang (Southwest University, China) for his important help in the
analysis of natural organic matter. Thanks are also due to Chen J., Kong K., Zhang Q.S. and Dr. Aslam M.W., for their
help with sample collection and measurements.
*Financial support.* This work was financially supported by the National Natural Science Foundation of China (41931297
and 42207164) and Guizhou Provincial Science and Technology Projects (No. Qian-Ke-He-Ji-Chu ZK [2022] Yi-Ban

566).



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
