# Peer review of "Dissolved organic matter fosters core mercury-methylating"

_EGUsphere, 2024_

## Community Comment (CC2)

**Reviewer Comments for the Manuscript:**

**Title:** Dissolved Organic Matter Fosters Core Mercury-Methylating Microbiome for Methylmercury Production in Paddy Soils

**Summary:** This manuscript presents an insightful study investigating the relationship between dissolved organic matter (DOM) and methylmercury (MeHg) production in paddy soils, particularly focusing on the core microbiome responsible for mercury methylation. The study employs advanced genomic and metagenomic techniques to identify key microbial taxa involved in mercury methylation and explores the impact of DOM on the activity of these microorganisms. The authors highlight that DOM plays a pivotal role in shaping the composition and functional activity of the Hg-methylating core microbiome, which in turn regulates MeHg production. The conclusions are supported by extensive field sampling, genome-resolved metagenomic analysis, and controlled incubation experiments.

**Strengths:**

1. **Scientific Significance**: The topic is timely and significant, particularly in the context of food safety and environmental pollution. Methylmercury contamination in rice, an essential staple food, is a critical issue.

2. **Methodological Rigor**: The use of hgcA gene sequencing combined with metagenomic analysis is an excellent approach to uncovering the microbial players in mercury methylation.

3. **Comprehensive Analysis**: The structural equation modeling and statistical approaches used provide a clear understanding of the relationship between Hg bioavailability, DOM, and Hg-methylating microbial communities.

4. **Practical Implications**: The study's insights on how human activities and climate change could influence MeHg formation rates through DOM variations have practical importance for environmental and public health policies.

**Weaknesses/Areas for Improvement:**

1. **Discussion Expansion**: The discussion section could further expand on how the findings integrate with broader environmental mercury cycling processes. A more detailed comparison with existing literature on Hg-methylating microbes in non-paddy soils would enhance the paper's contribution to microbial ecology.

2. **DOM Composition**: The manuscript discusses DOM composition but lacks specific details on the types of organic compounds most relevant to stimulating Hg-methylating activity. Including additional chemical analyses of DOM could strengthen the conclusions.

3. **Mechanistic Insights**: The authors hypothesize that DOM stimulates microbial activity through metabolic pathways but do not delve deeply into the

mechanistic underpinnings. Further discussion on the microbial metabolic pathways involved, possibly supported by metabolomic data, would improve the mechanistic understanding.

**Minor Comments:**

- The introduction is well-written but could benefit from a clearer explanation of the broader ecological importance of Hg methylation in paddy fields versus other environments.

- Line16 suggest "remains"

- Line 64 "…other factors also play roles in MeHg production…"

- Line 62, Line 237, Line 299, Lne 321 suggest unify the description of "low-molecular-weight DOMs" throughout the maus.

- Line 255 Whether the results support the argument that "MeHg concentration was strongly linked to hgcA gene abundance even compared to abiotic factors", it seems to be no description of the relevant results

**Recommendation:** The study provides valuable insights into the role of DOM in regulating mercury methylation in paddy soils. It is scientifically rigorous and offers practical implications for environmental health. I recommend the manuscript for publication after addressing the above-mentioned revisions.